# *Streptomyces* as Biofactories: A Bibliometric Analysis of Antibiotic Production Against *Staphylococcus aureus*

**DOI:** 10.3390/antibiotics14100983

**Published:** 2025-09-30

**Authors:** Pablício Pereira Cardoso, Kamila Brielle Pantoja Vasconcelos, Sámia Rocha Pereira, Rafael Silva Cardoso, Ramillys Carvalho de Souza, Lucas Francisco da Silva Nogueira, Suelen Fabrícia dos Santos Bentes, Vivaldo Gemaque de Almeida, Silvia Katrine Rabelo da Silva

**Affiliations:** 1Programa de Pós-Graduação em Recursos Naturais da Amazônia, Universidade Federal do Oeste do Pará—UFOPA, Santarém 68040-255, PA, Brazil; katrinerabelos@gmail.com; 2Instituto de saúde coletiva, Universidade Federal do Oeste do Pará—UFOPA, Santarém 68040-255, PA, Brazil; kamilabrielle@hotmail.com (K.B.P.V.); samiarocha6@gmail.com (S.R.P.); rafaelstm665@gmail.com (R.S.C.); lucasfranciscodasilvanogueira@gmail.com (L.F.d.S.N.); suellenbentes706@gmail.com (S.F.d.S.B.); 3Programa de Pós-Graduação em Ciências da Saúde, Universidade Federal do Oeste do Pará—UFOPA, Santarém 68040-255, PA, Brazil; 4Departamento de Microbiologia, Instituto de Ciências Biomédicas-ICB, Universidade de São Paulo—USP, São Paulo 05508-220, SP, Brazil; ramillyscarvalho@icb.usp.br; 5Programa de pós-graduação Ensino em Saúde na Amazônia (PPG-ESA), Universidade Estadual do Pará—UEPA, Belém 66087-670, PA, Brazil; vivaldo.gd.almeida@uepa.br

**Keywords:** MRSA, bioprospecting, genomics, secondary metabolites, antimicrobial resistance

## Abstract

Infections caused by *Staphylococcus aureus* pose significant public health challenges, particularly due to antibiotic-resistant strains like MRSA. In this context, *Streptomyces*, a genus known for producing natural antibiotics, emerges as a promising source for novel therapeutic agents. In this study, a bibliometric analysis of the scientific literature (2015–2024) on *Streptomyces* as antibiotic biofactories against *S. aureus* was performed, aiming to identify publication trends, collaborative networks, and emerging research areas. Using the Web of Science database, searches were performed with descriptors (“*Streptomyces*” AND “*Staphylococcus aureus*”), including original articles and reviews in English. Data were analyzed with VOSviewer and Biblioshiny to visualize collaborative networks, keyword co-occurrences, and trends. A total of 755 articles from 3705 authors were analyzed, highlighting significant collaboration (98.7%). Publications showed marked growth, particularly in Microbiology (21.7%), Pharmacology and Pharmacy (16.8%), and Biotechnology and Applied Microbiology (16.1%). China and India led in publication volume, whereas the United States exhibited the highest citation impact. Key emerging research topics include biosynthesis and metabolic optimization, antimicrobial activity and bioprospecting, mechanisms of antibiotic action and bacterial resistance, and genomic analyses. Research on *Streptomyces* for antibiotic production against *S. aureus* demonstrates continuous expansion and global interest, emphasizing the importance of international collaboration and multidisciplinary approaches. Future studies should intensify exploration of biodiverse environments, genetic engineering applications, and combinatorial strategies to effectively address antimicrobial resistance.

## 1. Introduction

*Staphylococcus aureus* infections constitute a critical public health challenge, exacerbated by the emergence of multidrug-resistant strains, particularly methicillin-resistant *S. aureus* (MRSA), which causes over 120,000 deaths annually worldwide [1,2]. This versatile pathogen is a leading cause of both nosocomial and community-acquired infections, manifesting as conditions ranging from superficial skin and soft tissue infections to invasive diseases such as pneumonia, bacteremia, and infective endocarditis, which are often refractory to conventional antimicrobial therapy [2,3,4].

Given the growing ineffectiveness of conventional antibiotics against resistant strains of *S. aureus*, a scientific consensus has been established regarding the urgent need to develop new antimicrobial agents with innovative mechanisms of action [2,5]. Contemporary research efforts prioritize natural sources for antimicrobial discovery, based on the proven track record of natural products in anti-infective chemotherapy and the urgent need to overcome bacterial resistance [6]. More than two-thirds of known natural antibiotics are biosynthesized by actinobacteria of the genus *Streptomyces*, consolidating this group as the primary microbial source of therapeutically relevant antimicrobial compounds [6,7].

Between 1981 and 2019, approximately 71% of clinically approved low-molecular-weight antibiotics were derived directly or indirectly from natural products, demonstrating the continued relevance of biodiversity as a source of antimicrobial pharmaceutical innovation [8], and a large portion of these compounds originate from *Streptomyces* metabolites [9]. These filamentous Gram-positive actinobacteria possess a diverse biosynthetic arsenal. They produce multiple classes of bioactive molecules with distinct mechanisms of action, including β-lactams, macrolides, and aminoglycosides [10].

Historically, *Streptomyces* have occupied a central position in combating staphylococcal infections. Their metabolites constitute cornerstones of modern antimicrobial therapy [11]. Clinically important antibiotics such as aminoglycosides (e.g., gentamicin) [12], macrolides (such as erythromycin) [13], and lipopeptides (like daptomycin) [14] originate from *Streptomyces* species.

The discovery of streptomycin in 1943 from *Streptomyces griseus* inaugurated the era of aminoglycosides. Subsequently, various *Streptomyces* species have provided effective molecules against *S. aureus*, including compounds capable of overcoming bacterial resistance mechanisms [15]. Even with the emergence of MRSA with expanded resistance, *Streptomyces* compounds maintain their therapeutic relevance. Daptomycin, a cyclic lipopeptide, constitutes an effective alternative to vancomycin for bacteremia and endocarditis caused by MRSA. Platensimycin represents another promising example of an anti-staphylococcal metabolite derived from this genus [14,16]. These examples highlight the ongoing importance of the *Streptomyces* genus as natural “biofactories” for potent antibiotics.

In the last decade, the prospection for new anti-*S. aureus* metabolites from *Streptomyces* has intensified significantly. This search is driven by the urgent need for therapeutic alternatives against multidrug-resistant strains [11]. Recent studies have reported the identification of new *Streptomyces* strains that biosynthesize metabolites with potent anti-staphylococcal activity, including compounds effective against multidrug-resistant MRSA and VRSA strains [17]. The isolation of the *Streptomyces* sp. YX44 strain exemplifies this trend, revealing broad-spectrum antimicrobial metabolites with proven efficacy against resistant strains of *S. aureus* [11]. Similarly, metabolites such as alnumycin D, obtained from *Streptomyces albus*, have demonstrated the ability to eradicate *S. aureus* biofilms at micromolar concentrations, targeting an important virulence factor of this pathogen [18].

Findings like these indicate that the chemical potential of *Streptomyces* is far from exhausted, paving the way for new classes of antibiotics or therapeutic adjuvants against *S. aureus* infections. At the same time, advances in genomics and biotechnology have made it possible to more broadly explore the potential of *Streptomyces* as cellular factories of bioactive molecules [19,20]. The analysis of complete *Streptomyces* genomes reveals that each species possesses between 20 and 50 gene clusters dedicated to the biosynthesis of secondary metabolites, of which up to ~90% remain “silent” under standard laboratory conditions [21]. This latent genetic richness suggests that many novel antibiotic molecules may still be discovered through the activation of cryptic biosynthetic pathways [21].

Modern strategies, including strain genetic engineering (“genetic mining”) and the heterologous expression of clusters in chassis hosts, have succeeded in awakening the production of hidden compounds and increasing the yields of known antibiotics [22]. Such approaches reinforce the view of *Streptomyces* as versatile biofactories: not only natural producers of antibiotics, but also organisms that can be optimized to produce greater quantities or improved variants of antimicrobial drugs [23]. Consequently, the research field involving *Streptomyces* and *S. aureus* has attracted growing interdisciplinary attention, bringing together microbiology, natural product chemistry, molecular biology, and pharmacology in the effort to discover and develop new therapies [24].

Considering this context, it becomes relevant to assess how the scientific community has been directing efforts to investigate *Streptomyces* in the search for anti-*S. aureus* agents. Bibliometric analyses emerge as an informative tool for mapping knowledge production in a given area, allowing the identification of trends, gaps, and key players involved. The main objective of the present study is to investigate the scientific output on the potential of *Streptomyces* as biofactories of antibiotics against *Staphylococcus aureus*, during the period from 2015 to 2024, based on data indexed in the Web of Science database. Specifically, it aims to achieve the following:Map the volume and temporal evolution of publications, as well as the citation index, highlighting growth trends in the field;Identify the most relevant journals, countries, institutions, and authors with the greatest influence and international collaboration, providing an overview of collaboration networks;Determine the most cited articles and the predominant keywords, highlighting emerging research lines and their respective contributions;Discuss the results in light of the growing challenges of bacterial resistance, in order to identify gaps and opportunities for the development of new antimicrobial therapies.

## 2. Materials and Methods

The bibliometric analysis was conducted to investigate research trends on *Streptomyces* as biofactories of antibiotics against *Staphylococcus aureus*, following established methodologies used in recent bibliometric studies [25,26]. The search strategy (Box 1) was designed to retrieve relevant articles from the Web of Science database (https://www.webofscience.com/, accessed on 1 March 2025), selected for its comprehensive coverage of the biological, pharmaceutical, and biotechnological sciences. Original articles and reviews written in English and published between 2015 and 2024 were included, provided they contained the descriptors “*Streptomyces*” AND “*Staphylococcus aureus*” in the titles, abstracts, or keywords. Subsequently, the authors manually reviewed the titles and abstracts of the publication set to eliminate irrelevant articles.

Box 1Search query sequence used in this study.(TS = (*Streptomyces*) AND (“*Staphylococcus aureus*”)) AND PUBLICATION YEARS: (2015–2024) AND (LIMIT-TO (LANGUAGE, “English”))

The bibliometric maps and collaboration networks in this study were constructed using two complementary software tools: VOSviewer (version 1.6.19) and Biblioshiny (version 4.1.2).

The analysis was configured with the following parameters: a minimum occurrence frequency of five for the nodes, full counting for co-occurrences, normalization by association strength, and the modularity-based clustering algorithm (Louvain). For the keyword networks, only terms with a minimum frequency of five appearances were selected.

The platform’s default parameters were used: a network based on a co-occurrence matrix, normalization by association strength, and automatic clustering using the Louvain modularity algorithm, with no additional frequency filters applied.

The metadata were exported in CSV format for analysis in the specified software. VOSviewer is a free Java-based tool developed for the construction and visualization of bibliometric maps with an intuitive graphical representation. Biblioshiny is a web interface of the R bibliometrix package (v 4.1.2) run on RStudio Desktop (v 2025.05.1-513, Posit Software, Boston, MA, USA) with R (v 4.5.1), providing comprehensive bibliometric analyses without the need for programming and facilitating data import, filtering, and diverse metric analyses.

Although each tool has distinct features in terms of functionality, the combination of both software platforms enables researchers to draw a broad overview of the evolution of a research field. This integrated approach considers indicators such as the quantitative growth of publications and citations, mapping of collaborative networks among countries, institutions, and researchers, as well as co-occurrence analysis of keywords [27].

## 3. Results

The main statistical information obtained from the bibliometric analysis is presented in Table 1. Although research on the genus *Streptomyces* is widespread, specific interest in the production of antibiotics against *Staphylococcus aureus* has shown significant growth in recent years. Between 2015 and 2024, a total of 755 original publications and review articles were identified, involving 3705 different authors, resulting in a total of 4776 author appearances. Of these, only 10 articles were written by a single author, while 745 involved multiple researchers, reinforcing the collaborative nature of this research area. Such articles were distributed across various international scientific journals, reflecting the global relevance of the topic and indicating strong interest in the discovery of new molecules with therapeutic potential in the face of the challenge of bacterial resistance.

After publication, each article was automatically categorized into different research areas in the Web of Science database. Figure 1 shows the percentage distribution of the main areas in which studies on the use of *Streptomyces* as biofactories of antibiotics against *Staphylococcus aureus* were conducted during the period from 2015 to 2024. The predominant areas were Microbiology (21.7%), Pharmacology and Pharmacy (16.8%), and Biotechnology and Applied Microbiology (16.1%). Combined, these three areas account for approximately 54.6% of the identified scientific production. Next, Medicinal Chemistry (12.5%) and Biochemistry and Molecular Biology (9.5%) stand out. Other important areas include Multidisciplinary Chemistry (6%), Plant Sciences (4.7%), Immunology (4.6%), Multidisciplinary Sciences (4.5%), and Organic Chemistry (3.5%). This broad disciplinary diversity highlights both the complexity and the interdisciplinary nature of the research on *Streptomyces* for antibiotic production, reflecting the plurality of methodological approaches and potential therapeutic applications.

Figure 2 presents the annual number of publications and accumulated citations on *Streptomyces* as biofactories of antibiotics against *Staphylococcus aureus* between 2015 and 2024. A significant increase in publications has been observed since 2015, starting with 51 articles and reaching a peak in 2022 with 94 published articles. The growth was particularly notable starting in 2017, which recorded 81 publications. Although there was a slight decrease to 71 articles in 2023, there was a subsequent recovery to 90 articles in 2024, reflecting a continuous and sustained interest in this research topic. This growth pattern reinforces the scientific relevance and the consistency of academic efforts to explore the antibiotic potential of the genus *Streptomyces* against *Staphylococcus aureus*, considering the growing global impact of bacterial resistance.

Figure 3 presents the number of citations per year of the articles published on the use of *Streptomyces* as biofactories of antibiotics against *Staphylococcus aureus*, highlighting a significant peak in the years 2017 and 2018, with 2697 and 2638 citations, respectively. After this period, a progressive reduction in the number of citations was observed, with 1196 citations recorded in 2019 and 941 in 2022. Starting in 2023, citations decreased considerably, with only 267 citations, which is expected due to the shorter time available since the publication of these articles to exert academic influence. This behavior is consistent with what is expected in the dynamics of scientific publications, where more recent articles tend to present an initially reduced number of citations, increasing gradually with the maturation and recognition of these publications by the scientific community.

Based on updated data, it is observed that China (147) and India (110) stand out as the countries with the highest number of published articles on the use of *Streptomyces* against *Staphylococcus aureus*, corresponding to 21.43% and 16.03% of the total publications, respectively. Following them are the USA (60; 8.75%), Egypt (47; 6.85%), and South Korea (40; 5.83%) (Table 2).

Regarding the number of citations, China (2341) and the USA (2064) appear as the most influential countries in the area, accounting for 15.40% and 45.90% of all citations, respectively. Although India is in second place in number of articles (110), it occupies the third position in terms of citations (1579; 13.30%). Egypt maintains fourth place in both categories, with 47 published articles and 1026 citations (25% of the total).

Next, South Korea (40 articles; 399 citations; 17.60%), Japan (33 articles; 475 citations; 17.60%), Germany (26 articles; 951 citations; 43.50%), Spain (24 articles; 383 citations; 23.90%), Thailand (20 articles; 96 citations; 4.80%), and France (18 articles; 120 citations; 15%) stand out, completing the list of the ten countries with the greatest scientific expression on this topic. These results highlight a wide geographical distribution and indicate the growing global and diversified collaboration in research related to the potential of *Streptomyces* in combating *Staphylococcus aureus*.

We mapped the cooperation of countries/regions using Bibliometrix. Cooperation occurred mainly among the United States, China, Asian countries, and European countries (Figure 4A). A visualization of the international collaboration network involving 64 countries that published at least a certain number of articles is illustrated in Figure 4B. Each node represents a country, and the size of the node is proportional to the volume of publications, while the thickness of the lines between the nodes reflects the intensity of their partnerships. It is observed that “peoples r China” and “India” appear in a central position in the network, indicating their prominent role as main contributors. In total, 13 different clusters are identified (highlighted by colors), formed from stronger collaborative links between certain groups of countries. Some blocks group European nations (e.g., Finland, Germany, and Poland), while others combine countries from Asia (such as Malaysia, South Korea, and Bangladesh), from the Middle East (Saudi Arabia, Iran), as well as partnerships involving countries from the Americas (Brazil, Canada, and Mexico), among others. At certain extremities of the map, we see more isolated nodes, such as “Russia” and “Switzerland”, suggesting somewhat less diversified collaborations. Overall, this representation provides a comprehensive view of the co-authorship landscape, highlighting both the most cooperative cores and the less frequent links among the 64 countries.

The ten most productive academics in research on the use of *Streptomyces* against *Staphylococcus aureus*, described in Table 3, were responsible for a total of 107 articles, representing approximately 14.17% of the publications identified in this bibliometric analysis. These researchers jointly accumulated 2169 citations, equivalent to 16.61% of the total citations received. Among these academics, researchers from Spain and China stand out, occupying four positions each, in addition to one representative from South Korea. Reyes, F. (258; 11.89%) and Zhang, Z.Z (290; 13.37%) occupied the top two positions, standing out not only for the number of articles published, but also for the significant quantity of citations received compared to the other authors. It should also be noted that Li, J. presented the highest h-index 3, indicating significant academic influence in his/her publications. The last positions among the top ten were occupied by researchers from Spain and China, including Martín, J. (154) and Li, Q.L (125), demonstrating the importance and geographical diversity of the most active researchers in this field.

A Table 4 presents the ten most cited articles on the use of *Streptomyces* as biofactories of antibiotics against *Staphylococcus aureus*, considering the total number of citations, the annual citation rate (TC per year), and the normalized number of citations (normalized TC). Standing out in first place is the study published by [28] in the journal *Frontiers in Microbiology*, which obtained a total of 573 citations and an annual rate of 71.63 citations, reflecting its significant impact in the field. In second place is the work by Dunbar et al. [29], published in *Chemical Reviews*, with 469 citations, indicating its relevance for the understanding of secondary metabolites produced by *Streptomyces*.

Also noteworthy are the articles by Smyth et al. [30], published in *Natural Product Reports*, and Martens et al. [31], in the *Journal of Antibiotics*, with 433 and 319 citations, respectively. These studies made fundamental contributions to the biosynthesis of antibiotics by actinobacteria and their application in combating resistant pathogens.

Furthermore, the analysis of the annual citation rate reveals the ongoing impact of the most recent publications. As an example, the article by [32] published in *Nanomaterials-Basel*, demonstrates high visibility, with an annual rate of 26.17 citations, reflecting the growing interest in the application of nanotechnological materials in enhancing the antimicrobial activity of compounds derived from *Streptomyces*.

The ranking presented in Table 4 therefore reinforces the international relevance of research on *Streptomyces* as an important source of new antibiotics, highlighting the growing impact of these studies in the face of the challenge posed by bacterial resistance.

**Table 4 antibiotics-14-00983-t004:** Most cited articles on *Streptomyces* against *Staphylococcus aureus*.

Paper	DOI	Total Citations	TC per Year	Normalized TC
Peterson, E.; Kaur, P. [28], *Front. Microbiol*. 2018	10.3389/fmicb.2018.02928	573	71.63	16.29
Dunbar, K.L. et al., [29], *Chem. Rev*. 2017	10.1021/acs.chemrev.6b00697	469	52.11	14.09
Smyth, J.E.; Butler, N.M.; Keller, P.A. [30], *Nat. Prod. Rep*. 2015	10.1039/c4np00121d	433	39.36	11.69
Martens, E.; Demain, A.L. [31], *J. Antibiot*. 2017	10.1038/ja.2017.30	319	35.44	9.58
Ashour, A.H. et al. [33], *Particuology* 2018	10.1016/j.partic.2017.12.001	241	30.13	6.85
Tang, X.; et al. [34], *ACS Chem. Biol*. 2015	10.1021/acschembio.5b00658	183	16.64	4.94
Taylor, S.D.; Palmer, M. [35], *Bioorgan. Med. Chem*. 2016	10.1016/j.bmc.2016.05.052	179	17.9	6.32
Allocati, N.; Masulli, M.; Di Ilio, C.; De Laurenzi, V. [36], *Cell Death Dis.* 2015	10.1038/cddis.2014.570	169	15.36	4.56
Salem, S.S. et al. [32], *Nanomaterials* 2020	10.3390/nano10102082	157	26.17	8.07
Miller, W.R.; Bayer, A.S.; Arias, C.A. [37], *Cold Spring Harb. Perspect. Med.* 2016	10.1101/cshperspect.a026997	151	15.1	5.33

Note: Total citations; TC per year = Total citations per year; Normalized TC = Time-normalized citation count.

Figure 5 presents the main journals that published studies on the use of *Streptomyces* against *Staphylococcus aureus*, revealing a significant concentration of publications in high-impact journals in the field of microbiology and natural products. Among the most relevant journals are the *Journal of Natural Products* (36 articles), *Frontiers in Microbiology* (35 articles), *Journal of Antibiotics* (30 articles), and *Marine Drugs* (29 articles), reflecting the growing interest of the scientific community in this line of research. The predominance of these journals indicates the relevance of the topic for the discovery of new bioactive compounds, especially in light of the growing concern with bacterial resistance. These data reinforce the importance of exploring the secondary metabolites produced by *Streptomyces*, with therapeutic potential for the development of innovative antibiotics.

The top ten institutions that published the most articles on the use of *Streptomyces* against *Staphylococcus aureus* are described in Table 5. The institution that leads in number of publications is the Egyptian Knowledge Bank (EKB), with a total of 121 articles. Next, the Chinese Academy of Sciences stands out with 78 publications, followed by Zhejiang University with 38 published articles. Other institutions with significant production include Kitasato University (29 articles), Seoul National University (SNU, 26 articles), and Sathyabama Institute of Science and Technology (25 articles). With a similar number of published articles are the Centre National de La Recherche Scientifique (CNRS, 24 articles), Council of Scientific and Industrial Research (CSIR) of India (24 articles), University of Chinese Academy of Sciences (CAS) (24 articles), and, finally, University of California System (21 articles). These data reinforce the international and collaborative character of research in this thematic area.

A word cloud with the main keywords identified in the articles related to the use of *Streptomyces* as biofactories of antibiotics against *Staphylococcus aureus* is presented in Figure 6. The words with higher frequency appear in proportionally larger size, indicating prominent relevance in the analyzed studies. Clearly, the terms “*Staphylococcus aureus*”, “antibiotics”, “biosynthesis”, “natural-products”, and “diversity” are the most prominent, highlighting the central focus of research in this area. This relevance confirms that the published studies predominantly focus on the identification, biosynthesis, and diversity of natural products obtained from *Streptomyces* with antibiotic activity, especially against the pathogen *S. aureus*, including resistant strains. Words like “antibacterial activity”, “identification”, “metabolites”, and “*Streptomyces* “ also emerge prominently, reflecting methodological aspects and predominant experimental approaches in the literature. The presented overview reveals important insights about current trends and the direction of studies focused on finding effective solutions against bacterial resistance through microbial biofactories.

A more detailed analysis using the co-occurrence network tool in VOSviewer (Figure 7) allowed visualizing how the keywords relate to each other, grouping them into four main clusters according to the frequency with which they appear simultaneously in the articles. Cluster 1 (34 terms) focuses on aspects related to the biosynthesis and optimization of metabolites (e.g., “biosynthetic gene-cluster”, “fermentation”, and “antibiotic production”), also including topics related to gene expression and structural characterization of *Streptomyces* compounds. Cluster 2 (33 terms), meanwhile, groups words associated with antimicrobial activity and bioprospecting (“antibacterial activity”, “antifungal activity”, and “metal nanoparticles”), highlighting usage strategies of extracts or nanoparticles in in vitro tests. In turn, Cluster 3 (30 terms) highlights mechanisms of action and bacterial resistance, featuring terms like “biofilm formation”, “mrsa”, and “antibiotic-resistance”, as well as target organisms such as *Bacillus subtilis*, *Escherichia coli*, and *Pseudomonas aeruginosa*. Finally, Cluster 4 (20 terms) groups references to the genomic study and taxonomic diversity of *Streptomyces* (“annotation”, “genome”, “strains”, and “taxonomy”), reinforcing the importance of molecular characterization in the development of new strategies to combat *Staphylococcus aureus*. Thus, the co-occurrence of keywords highlights a multidisciplinary research area, where the search for new antibiotics involves both strain engineering and the investigation of resistance mechanisms and biotechnological potential.

## 4. Discussion

The results of this bibliometric analysis reveal substantial and sustained growth in research on the application of *Streptomyces* against *Staphylococcus aureus* over the past decade. Between 2015 and 2024, a total of 755 publications related to this topic were identified, reflecting a significant volume of scientific activity. Only 10 articles (1.3%) were authored by a single researcher, while the vast majority (98.7%) were produced through multi-authored collaborations, reinforcing the premise that the discovery of new antibiotics relies on interdisciplinary teams and scientific cooperation [38].

The analysis of publication distribution by research area demonstrates an interdisciplinary profile, with a predominance in Microbiology (21.7%), Pharmacology and Pharmacy (16.8%), and Biotechnology and Applied Microbiology (16.1%). This multidisciplinary character reflects the inherent complexity of investigating *Streptomyces* as a source of antibiotics, which requires diverse approaches ranging from genomic analyses to antimicrobial activity assays. As noted by [39], this convergence of research areas has been fundamental in driving advances in the discovery of new bioactive metabolites.

The temporal growth of publications, starting from 51 articles in 2015 and reaching peaks such as the 94 articles in 2022, highlights the growing interest of the scientific community in the potential of *Streptomyces* as natural biofactories of antibiotics. This increase coincides with the worsening of global antimicrobial resistance, recognized by the World Health Organization (WHO) as one of the greatest threats to global health. Recent global reports estimate that infections by resistant *S. aureus* contributed significantly to mortality associated with antimicrobial resistance, totaling hundreds of thousands of deaths in 2019 [40].

The highest-impact articles reveal different perspectives on antibiotic production by *Streptomyces* and antimicrobial resistance. Peterson and Kaur [28], the most cited article with 573 citations published in *Frontiers in Microbiology*, examines the fundamental mechanisms of bacterial resistance and the relationships between resistance determinants in environmental bacteria and clinical pathogens. It emphasizes the need to understand how resistance mechanisms emerge and spread across different ecological contexts—knowledge that is essential for developing future therapeutic strategies.

Dunbar et al. [29] present a comprehensive review in Chemical Reviews with 469 citations, analyzing the enzymatic formation of carbon–sulfur bonds in the biosynthesis of natural products. They elucidate how these structural modifications influence the biological activity of compounds derived from actinobacteria, particularly *Streptomyces*, demonstrating their relevance to the discovery of effective antimicrobial agents against multidrug-resistant bacteria.

Smyth et al. [30], with 433 citations in *Natural Product Reports*, investigate the structural importance of atropisomers in biological systems, including secondary metabolites from *Streptomyces*, highlighting how the unique stereochemical properties of these compounds position them as promising candidates for pharmaceutical innovations, especially in antimicrobial therapies.

In a paper published in the *Journal of Antibiotics* with 319 citations, Martens and Demain [31] offer a detailed analysis of the global antimicrobial resistance crisis, focusing particularly on the United States. The work emphasizes the pivotal role of actinobacteria, especially *Streptomyces*, in discovering new antibiotics while also proposing strategies to overcome current challenges in the identification of bioactive compounds.

Finally, Ashour et al. [33] investigate in *Particuology*, with 241 citations, the use of metallic nanoparticles synthesized by sol–gel methods as antimicrobial agents, demonstrating how the combination of these nanoparticles with *Streptomyces* metabolites enhances antimicrobial efficacy against resistant pathogens, including *S. aureus*.

The geographical distribution of publications reveals a notable pattern, with China (147 articles; 21.43%) and India (110 articles; 16.03%) leading in total number of publications, followed by the United States (60 articles; 8.75%). However, when impact is assessed by citation count, the United States demonstrates a greater proportional influence, accumulating 2064 citations (45.90% citation share) compared to China’s 2341 citations (15.40%). This discrepancy between publication quantity and citation impact reflects differing levels of scientific maturity and international research integration—a phenomenon also observed in other emerging scientific fields [41].

The analysis of international collaboration networks revealed 13 distinct clusters, with China and India occupying central positions. This pattern of scientific cooperation is fundamental for the discovery of new antibiotics, as it facilitates the integration of diverse perspectives and resources. As exemplified by [42], multi-institutional collaborative projects have proven effective in identifying novel antimicrobial compounds, demonstrating the value of interdisciplinary interaction among microbiology, medicinal chemistry, and science education in addressing bacterial resistance.

Regarding the most productive researchers, scientists from Spain and China stand out, with Reyes F. and Zhang, Z.Z occupying prominent positions in number of publications and citations. This concentration of expertise in certain research groups reflects the necessary specialization for working with *Streptomyces*, which demands expertise in microbiology, genetics, and natural products chemistry. Examples of this are the studies by Reyes F., who collaborated in the discovery of new ikarugamycin derivatives with antifungal and antibacterial properties [43], and of Zhang, Z.Z, involved in the identification of the streptopyrazinones A–D, rare metabolites isolated from marine *Streptomyces* sp. ZZ446 [44].

Among the most cited articles, the study by Peterson et al. [28], with 573 citations, stands out for addressing fundamental aspects of antibiotic biosynthesis by actinobacteria. This high citation count demonstrates the enduring influence of seminal works in guiding subsequent research. As noted by Aria and Cuccurullo [25], highly cited articles in bibliometric analyses often establish methodological or conceptual paradigms that shape the evolution of a scientific field. For example, in the study by Polinko and Coupland [45] titled “Paradigm shifts in forestry and forest research: a bibliometric analysis”, the authors argue that changes in research trends reflect a paradigmatic transition towards sustainable forest management. Similarly, the analysis by [46] titled “Shifting research paradigms in landscape ecology: insights from bibliometric analysis” reveals that concepts such as “dynamics”, “disturbance”, and “landscape structure” were fundamental for the development of landscape ecology. These examples illustrate how influential articles can introduce new concepts or methodologies that reorient the trajectory of a field of study.

Among the most relevant journals, the *Journal of Natural Products* (36 articles), *Frontiers in Microbiology* (35 articles), and the *Journal of Antibiotics* (30 articles) stand out as leading publication venues. The predominance of these specialized journals underscores the focused and technical nature of research involving *Streptomyces*, highlighting the need for publication channels that address the specific requirements and nuances of this field.

The analysis of keyword co-occurrence revealed four main clusters that map the emerging subfields in research on *Streptomyces* against *S. aureus*. The first cluster, focused on biosynthesis and metabolic optimization, reflects the growing interest in the engineering of *Streptomyces* to increase antibiotic production or generate new derivatives. As reviewed by [47], strategies such as the activation of silent genes and regulatory manipulation have shown great potential to overcome production limitations in native strains.

The second cluster, focused on antimicrobial activity and bioprospecting, underscores the ongoing search for novel *Streptomyces* sources in previously unexplored environments. Recent studies conducted in the Brazilian Amazon have identified new *Streptomyces* lineages exhibiting potent activity against resistant *Staphylococcus aureus* [48]. This finding emphasizes the crucial role of investigating megadiverse ecosystems for the discovery of new antimicrobial agents.

Compounds derived from *Streptomyces* demonstrate diverse and highly specific mechanisms of action against *Staphylococcus aureus*, particularly against resistant strains such as MRSA. The bibliometric analysis revealed four main categories of antimicrobial action: (1) inhibition of cell wall synthesis, exemplified by β-lactams and glycopeptides that interfere with peptidoglycan formation; (2) interference with protein synthesis, a characteristic of aminoglycosides (like gentamicin) and macrolides (like erythromycin), which bind to bacterial ribosomes; (3) direct action on the cell membrane, represented by daptomycin, a lipopeptide that binds to the membrane in the presence of calcium, causing depolarization and inhibition of nucleic acid synthesis [35,37]; and (4) inhibition of specific metabolic processes, as observed with platensimycin, which inhibits the FabF enzyme in fatty acid biosynthesis [16], and granaticin, which demonstrates a unique ability to inhibit biofilm formation, a major virulence factor of *S. aureus* [49]. These varied mechanisms explain the continued efficacy of *Streptomyces* metabolites against resistant strains and justify the growing scientific interest in exploring this genus as a source of novel therapeutic agents with a low risk of cross-resistance.

The third cluster, focused on mechanisms of action and resistance, reflects the increasing emphasis on understanding factors that affect antimicrobial efficacy against resistant strains, particularly methicillin-resistant *Staphylococcus aureus* (MRSA). Recent studies, such as [49], have demonstrated that certain metabolites produced by *Streptomyces*—including granaticin derivatives identified in strain QHH-9511—exhibit significant activity against MRSA and possess the ability to inhibit *S. aureus* biofilms, a key virulence factor of this pathogen [49].

The fourth cluster, focused on genomics and taxonomic diversity, reflects the advances in sequencing technologies that have revolutionized the exploration of the biosynthetic potential of *Streptomyces*. Recent genomic analyses, such as those by [21], revealed that each *Streptomyces* species can contain between 20 and 50 gene clusters dedicated to the biosynthesis of secondary metabolites, of which a large part remains “silent” under standard laboratory conditions, indicating a vast unexplored potential for the discovery of new antibiotics. This vast unexplored genetic reservoir represents a promising frontier for the discovery of new antibiotics.

An important limitation of this bibliometric study was the restriction to a single database (Web of Science), which may have resulted in the omission of publications indexed exclusively in other platforms. Furthermore, as observed by [50], bibliometric analyses capture quantitative trends in scientific production, but do not necessarily reflect the practical impact of this research on the development of new commercially viable antimicrobials.

The international collaboration clusters identified in this study reveal significant implications for global scientific output. The observed predominance of North–South partnerships presents a dual effect. Specifically, it facilitates the transfer of scientific and technical knowledge and financial resources to developing countries, but at the same time, it can perpetuate relations of scientific and technological dependency among nations with different levels of development.

The predominance of English as the language of publication introduces a linguistic bias that potentially excludes relevant scientific contributions produced in other languages, limiting a comprehensive understanding of the field. Additionally, restricted access to scientific databases in regions with limited resources compromises the international visibility of local scientific production, resulting in an unequal representation in the global literature on *Streptomyces* as antibiotic producers.

Despite these limitations, the results obtained provide a comprehensive overview of the current state of research on *Streptomyces* as biofactories of antibiotics against *S. aureus*. In a scenario of growing antimicrobial resistance, with projections estimating up to 10 million annual deaths by 2050, studies such as that by [51] demonstrate the enormous genomic potential of these microorganisms for the discovery of antibiotics with a low risk of resistance, reinforcing their strategic role in confronting this global challenge.

## 5. Conclusions

The bibliometric analysis showed continuous growth in publications on *Streptomyces* as biofactories of anti-*Staphylococcus aureus* antibiotics (2015–2024), with peaks in 2022 and 2024. The high level of co-authorship (98.7%) reflects international collaborations structured into 13 distinct clusters, with China and India standing out in terms of production volume and the United States in terms of proportional impact. The co-occurrence analysis identified four emerging sub-areas: biosynthesis and metabolic optimization, antimicrobial activity and bioprospecting, mechanisms of action and resistance, and genomics and taxonomic diversity.

To tackle growing antimicrobial resistance, it is recommended to intensify bioprospecting in megadiverse environments such as the Brazilian Amazon, employ genetic engineering to activate silent gene clusters, and develop combinatorial strategies using *Streptomyces* metabolites as therapeutic adjuvants. The results reveal a dynamic field with significant potential for innovative solutions to contemporary antimicrobial challenges.

## Figures and Tables

**Figure 1 antibiotics-14-00983-f001:**
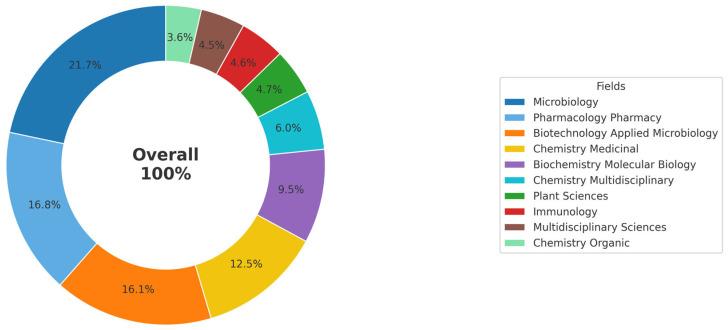
Percentage distribution of research areas for publications on *Streptomyces* as biofactories of antibiotics against *Staphylococcus aureus* (2015–2024). The figure highlights the predominance of the fields of Microbiology, Pharmacology and Pharmacy, as well as Biotechnology and Applied Microbiology, reflecting the interdisciplinary nature and complexity of research on this topic.

**Figure 2 antibiotics-14-00983-f002:**
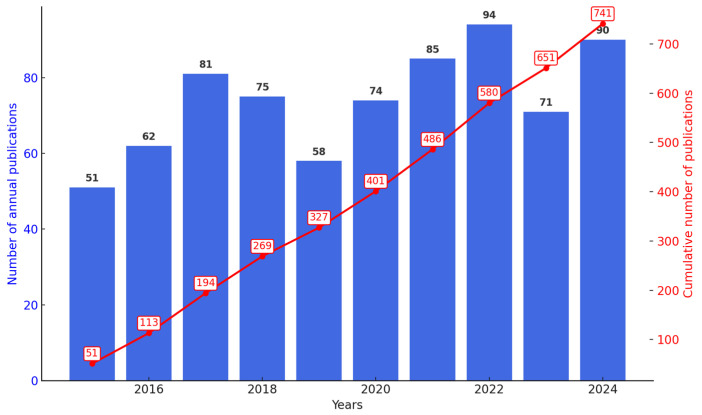
Annual and cumulative number of scientific publications on the use of *Streptomyces* to combat *Staphylococcus aureus* from 2015–2024. A clear upward trend is observed, especially since 2017, indicating a continuous and significant interest from the scientific community in this area.

**Figure 3 antibiotics-14-00983-f003:**
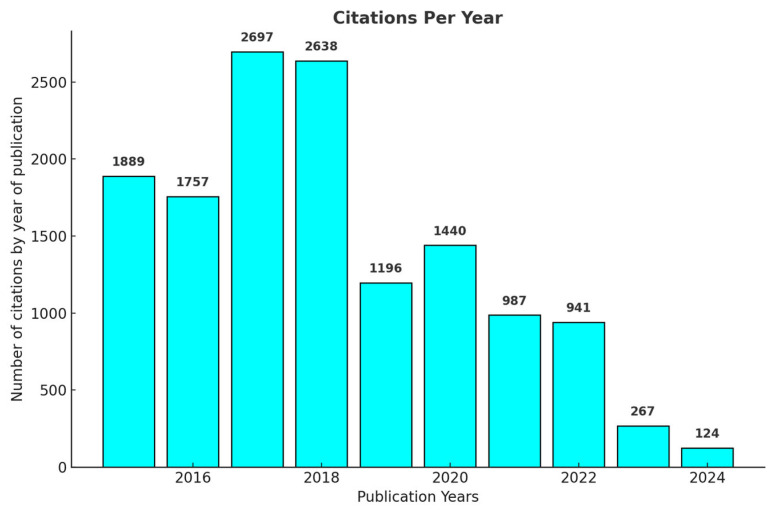
Number of citations received annually by articles published on the use of *Streptomyces* as producers of antibiotics against *Staphylococcus aureus* from 2015–2024. The figure shows a peak in citations in the first few years after publication, particularly in 2017 and 2018, followed by a gradual decrease.

**Figure 4 antibiotics-14-00983-f004:**
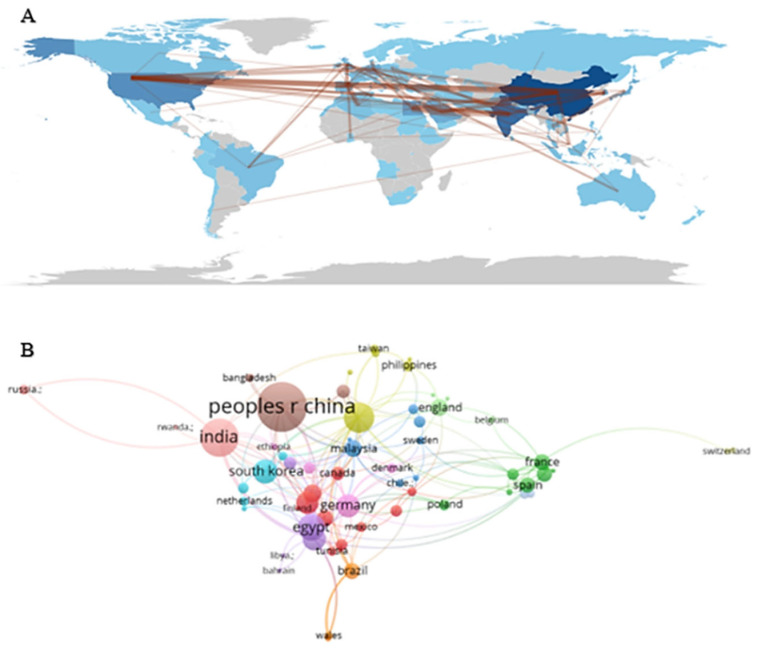
Global mapping (**A**) and international collaboration network (**B**) among countries that published research on *Streptomyces* as antibiotic biofactories against *S. aureus*. The world map visually shows the relative volume of publications by country, while the bibliometric network reveals specific relationships between nations, organized into 13 distinct clusters. China and India stand out as central hubs in international collaborations, reinforcing the leading role of these regions in natural product-based antimicrobial research.

**Figure 5 antibiotics-14-00983-f005:**
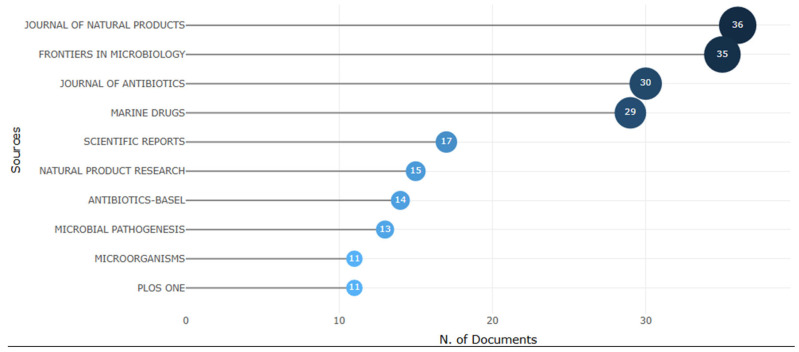
Leading scientific journals that have published articles related to the use of *Streptomyces* against *Staphylococcus aureus*. The figure highlights renowned and specialized journals, such as the Journal of Natural Products, Frontiers in Microbiology, Journal of Antibiotics, and Marine Drugs, indicating that these periodicals are considered the main channels for the dissemination of research related to the discovery of new antibiotic compounds.

**Figure 6 antibiotics-14-00983-f006:**
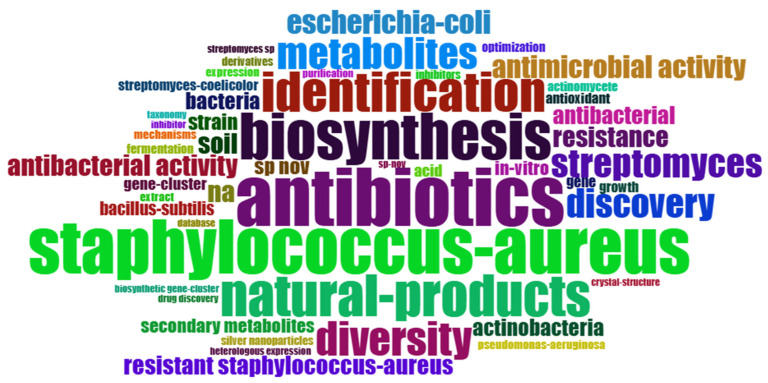
Word cloud of the most frequent keywords identified in the analyzed articles on the use of *Streptomyces* as antibiotic biofactories against *S. aureus*. The highlighted words are proportional to their frequency, indicating that themes such as ‘*Staphylococcus aureus*’, ‘antibiotics’, ‘biosynthesis’, ‘natural products’, and ‘diversity’ are at the center of the analyzed studies. This figure provides a clear visual representation of the main scientific and methodological approaches that characterize research in the area, allowing for the rapid identification of the most relevant and addressed topics in recent literature. Note: The size of the words reflects their relative frequency in the analyzed literature.

**Figure 7 antibiotics-14-00983-f007:**
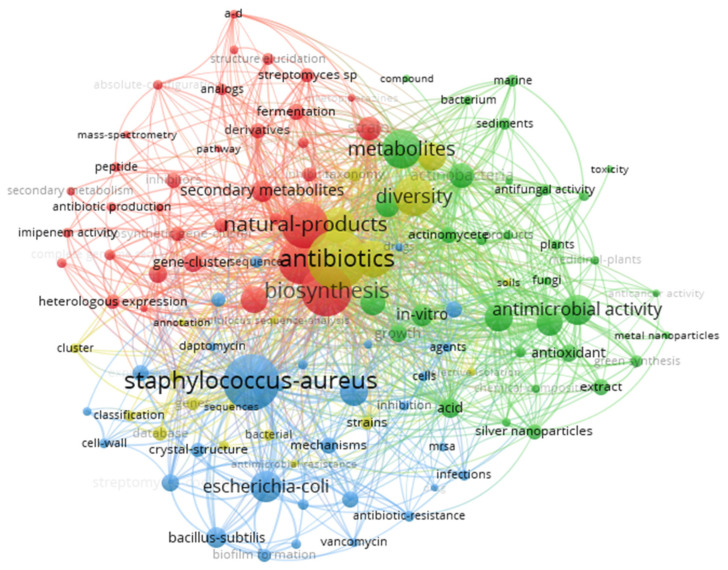
A bibliometric network showing the co-occurrence of the most frequent keywords from articles on *Streptomyces* against *Staphylococcus aureus* (2015–2024). In this network, each node represents a keyword, and its size is proportional to its frequency within the analyzed literature. The connecting lines (or edges) illustrate the co-occurrence strength, indicating how frequently keywords appear together in the same document. The colors correspond to four distinct thematic clusters identified by the analysis: Cluster 1 (red) is related to biosynthesis and metabolic production (e.g., biosynthesis, fermentation, antibiotic production, gene cluster, regulation); Cluster 2 (green) covers antimicrobial activity and bioprospecting (e.g., antibacterial activity, antifungal activity, nanoparticles, extract, inhibition); Cluster 3 (blue) deals with mechanisms of action and bacterial resistance (e.g., MRSA, resistance, biofilm formation, multidrug resistance, Escherichia coli, Bacillus subtilis); and Cluster 4 (yellow) encompasses terms concerning genomic and taxonomic diversity (e.g., genome, annotation, taxonomy, strain identification, sequencing).

**Table 1 antibiotics-14-00983-t001:** General statistical information on the scientific production related to the use of *Streptomyces* as antibiotic biofactories against *Staphylococcus aureus* published in the Web of Science database in the period from (2015–2024).

Description	Value
Original articles and review articles	755.0
Authors	3705.0
Appearances of the author	4776.0
Authors of single-author documents	10.0
Authors of multi-author documents	746.0
Authors per document	6.32
Co-authors per document	5.32
Documents per author	0.2

**Table 2 antibiotics-14-00983-t002:** The ten most productive countries based on the number of articles and citations.

Order	Country	Total Articles	% Articles	Total Citations	% Citations
1	China	147	21.43	2341	15.4
2	India	110	16.03	1579	13.3
3	USA	60	8.75	2064	45.9
4	Egypt	47	6.85	1026	25
5	South Korea	40	5.83	399	17.6
6	Japan	33	4.81	475	17.6
7	Germany	26	3.79	951	43.5
8	Spain	24	3.5	383	23.9
9	Thailand	20	2.92	96	4.8
10	France	18	2.62	120	15

**Table 3 antibiotics-14-00983-t003:** Top ten academics based on the number of articles.

Order	Author	Institution/Country	Nº of Articles	Nº of Citations
1	Reyes, F.	Fdn Medina, Ctr Excelencia Invest Medicamentos Innovadores An, Granada 18016/Spain	14	258
2	Zhang, Z.Z.	Zhejiang Univ, Ocean Coll, Zhoushan Campus, Zhoushan 316021/China	13	290
3	Ju, J.H.	Chinese Acad Sci, Cas Key Lab Trop Marine Bioresources & Ecol/China	12	218
4	Li, J.	Shanghai Jiao Tong Univ, Ren Ji Hosp, Res Ctr Marine Drugs, Dept Pharm, State Key Lab Oncogenes & Related Genes, Sch Med/China	11	361
5	Lian, X.Y.	Zhejiang Univ, Ocean Coll, Zhoushan Campus, Zhoushan 316021/China	11	242
6	Lee, J.	Seoul Natl Univ, Coll Agr & Life Sci, Dept Agr Biotechnol, Seoul 08826/South Korea	10	116
7	Genilloud, O.	Fdn Medina, Ctr Excelencia Invest Medicamentos Innovadores An, Granada 18016/Spain	9	184
8	Li, Q.L.	Chinese Acad Sci, Cas Key Lab Trop Marine Bioresources & Ecol/China	9	125
9	Martín, J.	Fdn Medina, Ctr Excelencia Invest Medicamentos Innovadores An, Granada 18016/Spain	9	154
10	Vicente, F.	Fdn Medina, Ctr Excelencia Invest Medicamentos Innovadores An, Granada 18016/Spain	9	221

**Table 5 antibiotics-14-00983-t005:** Main institutions in number of scientific publications.

Affiliation	Country	Articles
Egyptian Knowledge Bank (EKB)	Egypt	121
Chinese Academy of Sciences	China	78
Zhejiang University	China	38
Kitasato University	Japan	29
Seoul National University (SNU)	South Korea	26
Sathyabama Institute of Science and Technology	India	25
Centre National de La Recherche Scientifique (CNRS)	France	24
Council of Scientific and Industrial Research (CSIR)—INDIA	India	24
University of Chinese Academy of Sciences (CAS)	China	24
University of California System	United States	21

## Data Availability

The data presented in this study are available within the article. Raw data supporting this study are available from the corresponding author upon reasonable request.

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
