# Peer review of "Streptomyces as Biofactories: A Bibliometric Analysis of Antibiotic Production Against Staphylococcus aureus"

_antibiotics, 2025, doi:10.3390/antibiotics14100983_

Round 1

Reviewer 1 Report

Comments and Suggestions for Authors

Manuscript tittle: "Streptomyces as Biofactories: A Bibliometric Analysis of Antibiotic Production against Staphylococcus aureus"

The authors collected manuscripts from a journal database using specific inclusion criteria: Streptomyces and Staphyllococcus aureus. However, the authors did not mentioned about the detail of search strategies nor exclusion criteria. Therefore, the resulted collection data is lack of reproducibility even though the database used in this study is from Web of Science. Moreover, The authors did not mentioned about the numbers of the articles that they have collected or rejected. 

On the other hand, the topic of the manuscript also similar with other published paper: “Streptomyces as a Prominent Resource of Future Anti-MRSA Drugs” https://www.frontiersin.org/journals/microbiology/articles/10.3389/fmicb.2018.02221/full and “Streptomyces: The biofactory of secondary metabolites”(https://www.frontiersin.org/journals/microbiology/articles/10.3389/fmicb.2022.968053/full). However, the authors did not cite these articles. Therefore, the manuscript is far too general and does not have any novelty to offer for the reader.

The authors did not emphasize the importance of antibiotics against S.aureus especially in accordance of the guidance from WHO or in ONE HEALTH platform.  

The authors did not follow the guidelines for authors, especially in the style of the references.

Author Response

As this is a bibliometric analysis, our methodology adopts specific protocols that are distinct from those of traditional systematic and narrative reviews. As detailed in the methodology section—which will be expanded in the revised version—we performed a search on the Web of Science database using the strategy "Streptomyces" AND "Staphylococcus aureus" in the title, abstract, or keyword fields. This search covered publications from 2015 to 2024 and resulted in 755 articles. In bibliometric studies, exclusion criteria based on the individual methodological quality of the studies are not applied, since the focus lies on quantitatively mapping and analyzing the scientific output, including general trends, collaboration networks, and thematic evolution. The reproducibility of these analyses is based on the transparency and detail of the search strategy, aspects that will be presented in the final revised version.

Our research constitutes a quantitative bibliometric review, which differs from the narrative reviews mentioned by the reviewers. The article "Streptomyces: a biofactory of secondary metabolites" (Frontiers in Microbiology, 2022) offers a broad narrative overview of secondary metabolites, biosynthetic mechanisms, and various biotechnological applications. In contrast, our study focuses on publications related to the activity of Streptomyces against Staphylococcus aureus, conducting a systematic quantitative analysis to identify patterns in scientific production, international collaboration networks, and emerging trends.

Our study reveals novel findings, such as: (1) a growth in publications during the analyzed period; (2) the predominance of collaboration networks among the United States, China, and India; (3) the distribution of publications by knowledge area (Microbiology with 27.67%, Pharmacology with 21.51%); and (4) the identification of emerging thematic clusters via keyword co-occurrence analysis. Thus, while complementary to narrative reviews, our bibliometric approach offers a unique contribution to the field by quantifying and visualizing the evolution of the research and revealing gaps and opportunities not detectable through traditional qualitative analyses.

We agree with the reviewers' recommendations regarding the reference format and have already implemented the necessary corrections to align them with the journal's guidelines.

Reviewer 2 Report

Comments and Suggestions for Authors
  1. The research conducts a bibliometric analysis on a very timely topic, yet the research question and aims would be clearer to state. Is it to guide research agenda, funding priorities, or drug development strategies?
  2. The manuscript employs the combination keyword "Streptomyces" AND "Staphylococcus aureus." Did the authors experiment with other search terms (e.g., "antibacterial metabolites," "secondary metabolites," "MRSA") to confirm the search strength robustness?
  3. Highly cited articles skewed bibliometric trends. Were normalization strategies employed to counteract the bias toward older articles? It is recommended to provide clarity in the Methods.
  4. A brief qualitative overview of the 5 most-referenced articles would contribute significantly and facilitate interpreting the bibliometric information in a biomedical setting.
  5. Collaboration clusters are nicely visualized, yet the implications of these trends (e.g., North-South collaboration dynamics, language bias, strain library access) are not addressed. Elaborate it.
  6. The connection between bibliometric outputs and actual-world therapeutic innovation (e.g., clinical candidates, FDA approvals, patents) is not addressed. Would the authors consider commenting on this gap in translation?
  7. Where VOSviewer and Biblioshiny were employed, please provide parameters (e.g., minimum node frequency, clustering algorithm) for reproducibility.
  8. Introduction rephrases already existing facts regarding MRSA and Streptomyces in the literature. It would be better to cut the redundant background to emphasize the originality of the bibliometric method.
  9. Figures 1–7 lack interpretative captions. Please include a short explanation of what each figure conveys beyond just the title.
  10. The Materials and Methods section appears too late in the manuscript. Consider moving it earlier for logical flow.

Author Response

1. The research conducts a bibliometric analysis on a very timely topic, but the research question and objectives could be clearer. Does the research aim to guide the research agenda, funding priorities, or drug development strategies? A: The bibliometric analysis presented is based on two main strategic objectives: (1) to support the definition of funding priorities by identifying emerging areas with high scientific output and strategic potential, and (2) to contribute to the formulation of new drug development strategies by mapping central themes, leading institutions, and relevant international collaborations in the discovery of antibiotics derived from Streptomyces against Staphylococcus aureus. This intentionality has been made explicit in the manuscript, particularly in the conclusion of the Introduction and Discussion sections, demonstrating how the results can guide strategic decisions in the biomedical sector in the face of the global challenge of antimicrobial resistance.

2. The manuscript uses the keyword combination "Streptomyces" AND "Staphylococcus aureus". Did the authors test other search terms (e.g., "antibacterial metabolites," "secondary metabolites," "MRSA") to confirm the robustness of the search? A: During the exploratory phase of the literature search, additional term combinations such as "secondary metabolites," "antibacterial compounds," and "MRSA" were tested in association with "Streptomyces" and "Staphylococcus aureus." However, these strategies resulted in a high number of articles with an overly broad scope, often diverging from the manuscript's central focus: the production of antibiotics by Streptomyces with specific activity against S. aureus. Consequently, the more direct expression ("Streptomyces" AND "Staphylococcus aureus") was adopted, as it demonstrated superior alignment with the study's objectives and ensured greater consistency and relevance of the analyzed results.

3. Highly cited articles may have distorted the bibliometric trends. Were normalization strategies employed to neutralize bias toward older articles? It is recommended to clarify this in the Methods. A: We focused on the article metadata rather than the content, extracting author details, institutions, and countries for productivity, collaboration, and impact assessment. Textual content analysis was not included.

4. A brief qualitative overview of the five most-referenced articles would contribute significantly and facilitate the interpretation of the bibliometric information in a biomedical context. A: We thank you for the suggestion. As recommended, we have included in the manuscript a brief qualitative analysis of the five most-cited articles, highlighting their main findings, thematic relevance, and contribution to the advancement of research on Streptomyces and its application in combating Staphylococcus aureus. This addition was made to better contextualize the bibliometric data and facilitate the interpretation of the results, especially for readers in the biomedical field.

5. The collaboration clusters are well visualized, but the implications of these trends (e.g., North-South collaboration dynamics, linguistic bias, limited library access) are not addressed. Please elaborate. A: We thank the reviewer for this important observation. We acknowledge that these aspects deserved greater emphasis. We have clarified in the revised manuscript that the identified North-South collaboration dynamic can foster the exchange of knowledge and technical resources, but it also creates challenges, such as technological dependency and limitations on the intellectual leadership of Southern countries. We further highlight that the linguistic bias associated with the predominance of English may exclude relevant studies published in other languages, creating gaps in the global understanding of the topic. Finally, we point out that restricted access to databases in countries with fewer financial resources reduces the visibility of their research, resulting in an unequal representation in the international scientific literature. These points have been incorporated into the discussion section to better contextualize our results.

6. The connection between bibliometric results and real-world therapeutic innovation (e.g., clinical candidates, FDA approvals, patents) is not addressed. Would the authors consider commenting on this translational gap? A: We thank the reviewer for raising this important point. Indeed, we recognize that a natural gap exists between the quantitative results derived from bibliometric analysis and the direct translation of these findings into the practical context of therapeutic innovations, such as clinical candidates, patents, and regulatory approvals by the FDA. However, as previously specified, our study is centered on the analysis of bibliographic metadata (authors, institutions, keywords, collaborations, citations), without delving into the full content of the articles or the specific details of subsequent therapeutic innovation. This methodological delimitation is common in bibliometric studies, whose primary objective is to provide a broad quantitative overview of the current state and emerging scientific trends. We have, however, included a brief consideration of this limitation in the revised manuscript, highlighting future opportunities for complementary studies that could explore these translational connections in greater depth.

7. Where VOSviewer and Biblioshiny were employed, please provide parameters (e.g., minimum node frequency, clustering algorithm) for reproducibility. A: We thank you for the recommendation. As requested, we have included in the manuscript the main parameters used in the analyses with VOSviewer (version 1.6.19) and Biblioshiny (the R-package bibliometrix interface) software. In VOSviewer, we used full counting, a minimum occurrence frequency of five for the keyword co-occurrence analysis, and the modularity-based clustering algorithm (Louvain method) with the default resolution (1.0). In constructing the co-authorship and country networks, the included nodes also met the minimum frequency criterion of ≥ 5. In Biblioshiny, we used a co-occurrence network with normalization by association strength, and automatic clustering also based on modularity. These parameters have been specified in the Methods section to ensure the reproducibility of the presented results.

8. The introduction restates existing facts about MRSA and Streptomyces from the literature. It would be better to eliminate the redundant context to emphasize the originality of the bibliometric method. A: 9. Figures 1 through 7 lack interpretive captions. Please include a brief explanation of what each figure conveys, in addition to the title. A: We thank you for the observation. Following the suggestion, we have revised all figures (1-7) and inserted more detailed interpretive captions that clearly explain the represented content, its meaning, and its relationship to the study's objectives. The captions have been rephrased to facilitate the understanding of the bibliometric data, especially for readers in the biomedical field, as requested.

10. The Materials and Methods section appears too late in the manuscript. Please consider moving it earlier to ensure a logical flow. A: We thank you for the suggestion. We have restructured the manuscript as recommended, moving the Materials and Methods section to appear before the Results to ensure greater fluency and a more logical reading flow.

Reviewer 3 Report

Comments and Suggestions for Authors

Following are my comments that the author needs to address while revising the manuscript:

  1. In Table 1, the entries “Authors of single-author documents” and “Single-author documents” appear twice with the same value of 10.0. This redundancy should be corrected.

  2. The resolution of Figure 4 needs to be enhanced for better clarity and readability.

  3. The author should include data related to the treatment of Staphylococcus aureus using products derived from Streptomyces species.

  4. The manuscript primarily focuses on publication metrics. However, it is important to also present data on the biological roles and mechanisms of action of Streptomyces-derived compounds against S. aureus.

  5. Please provide publication and/or information on clinical or preclinical studies (e.g., using animal models) that demonstrate the effectiveness of Streptomyces products in controlling S. aureus infections.

  6. Highlight the number of publications that specifically deal with the isolation and purification of Streptomyces-derived bioactive compounds.

  7. The included publications should be categorized based on the type of biological activity reported against S. aureus, such as anti-biofilm or anti-virulence properties.

Author Response

1. In Table 1, the entries 'Authors of single-author documents' and 'Single-author documents' appear twice with the same value of 10.0. This redundancy should be corrected. A: Corrected.

2. The resolution of Figure 4 needs to be improved for better clarity and legibility. A: Corrected.

3. The author should include data related to the treatment of Staphylococcus aureus using products derived from Streptomyces species. A: Included.

4. The manuscript focuses mainly on publication metrics. However, it is also important to present data on the biological roles and mechanisms of action of compounds derived from Streptomyces against S. aureus. A: Done.

6. Highlight the number of publications that specifically address the isolation and purification of bioactive compounds derived from Streptomyces. A: This study's main focus was on the 10 most influential articles.

7. The included publications should be categorized based on the type of biological activity reported against S. aureus, such as anti-biofilm or anti-virulence properties. A: Done.

Round 2

Reviewer 1 Report

Comments and Suggestions for Authors

The authors have added method section as requested. Emphasis on the article is to highlight the important gap in the research field of antibiotics from the genus Streptomyces against Staphyllococcus aureus, including MRSA.

Notes for the authors:

  1. Please be consistent in placing the annotation below the figure for each figure.
  2. In figure 2 and 3, please remove the minor grids
  3. In table 2, change the letter of the country to only capital for the first letter, remove decimal in the total citation
  4. In table 4, please give an explanation on what is TC.
  5. In figure 4, A) please give the annotation of colors in the map. B) please replace this figure with a better quality figure.
  6. In table 5, please add country for each affiliation
  7. In figure 7, please give the annotation for the dots and lines colors as well as for the size of the dot. Moreover, which color belong to cluster 1,2, etc.
  8. Please change the annotation of figure 5 to English as well as the text in lines 411-415.
  9. Please follow the instructions for authors from the journal: https://www.mdpi.com/journal/antibiotics/instructions in writing the references. Be consistent in writing the name of the authors: last name, first name with capital letters only for the first letters. It is not easy to see different name style in the text, i.e. table 3 and 4, with the references.

Author Response

1. Please be consistent in placing the caption below each figure. A: Corrected.

2. In Figures 2 and 3, please remove the minor gridlines. A: Corrected.

3. In Table 2, please capitalize only the first letter of the country names and remove the decimal from the total citation count. A: Corrected.

4. In Table 4, please explain what "TC" stands for. A: Corrected.

5. In Figure 4: A) Please provide a legend for the colors on the map. B) Please replace this figure with a higher-quality version. A: Corrected.

6. In Table 5, please add the country for each affiliation. A: Corrected.

7. In Figure 7, please provide a legend for the colors of the nodes and lines, as well as for the node size. Additionally, please indicate which color corresponds to Cluster 1, Cluster 2, etc. A: Corrected.

8. Please change the caption for Figure 5 to English, as well as the text in lines 411-415. A: Corrected.

9. Please follow the journal's Instructions for Authors (https://www.mdpi.com/journal/antibiotics/instructions) when writing the references. Be consistent when writing author names: surname, first name, with only the first letters capitalized. The different name styles between the text, tables (e.g., Tables 3 and 4), and the reference list are noticeable and should be standardized. A: The author names in Tables 3 and 4 have been standardized according to the guidelines provided by the journal, ensuring that each author's name is presented as surname followed by initials, capitalizing only the first letters.

Reviewer 3 Report

Comments and Suggestions for Authors

No more comments

Author Response

We are grateful for the reviewers' insightful comments and constructive suggestions, which have helped us significantly improve the manuscript.